# Paraffin–Peloid Formulations from Copahue: Processing, Characterization, and Application

**DOI:** 10.3390/ma16145062

**Published:** 2023-07-18

**Authors:** Micaela A. Sanchez, Miria T. Baschini, Manuel Pozo, Betina R. Gramisci, María E. Roca Jalil, María L. Vela

**Affiliations:** 1Engineering School, Comahue National University, Neuquén 8300, Argentina; 2Institute for Research and Development in Process Engineering, Biotechnology and Alternative Energies (PROBIEN), CONICET-Comahue National University, Neuquén 8300, Argentina; 3Department of Geology and Geochemistry, Faculty of Sciences, Autonomous University of Madrid, Campus de Cantoblanco, 28049 Madrid, Spain; 4Health and Environment Sciences School, Comahue National University, Neuquén 8300, Argentina

**Keywords:** mudpack, thermotherapy, dry peloids, paraffin

## Abstract

The Copahue Thermal Center, situated in Neuquén, Argentina, produces natural and matured peloids, which are employed in the prevention and treatment of various osteoarticular and dermatological disorders. The presence of sulfur as a constituent and its thermotherapeutic potential constitute the primary strengths of these peloids. Nevertheless, accessing Copahue is challenging due to its distance from densely populated centers and the snow cover during the winter months in the southern hemisphere. Therefore, in order to propose a material that can be utilized year-round in any location, a mudpack was obtained by combining medicinal paraffin with dehydrated Copahue peloids, with concentrations evaluated up to 10% *w/w*. This mudpack was analyzed through X-ray diffraction, which detected the presence of sulfur, the most important component of Copahue’s peloids. Through IR spectroscopy, the signals that identify medicinal paraffin were clearly observed, and for concentrations of 6% and 10% peloid in the material, it was possible to detect the presence of mineral clay components associated with Si-O stretching vibrations at around 1041 cm^−1^. The low values of luminosity and grey tonality obtained for the mudpack contributed to patient acceptability and the absorption of electromagnetic radiation. The experimental cooling rate, calculated using the ratio of the temperature variation (∆T) with respect to the time variation (∆t) in each interval of the experimental curve, was determined to be 0.6 °C·min^−1^ for both paraffin and the mudpack. However, for peloids, higher values ranging from 0.6 to 4.8 °C·min^−1^ were obtained. This suggests that the mudpack mixtures have a slower heat release, which is a desirable property for their use as a thermotherapeutic agent. Considering the reusability of the mudpack, its stability was evaluated after 10 cycles of cooling and heating through XRD, DSC, and FTIR tests, resulting in a system that retains its properties. The formulation of the obtained mudpack is promising for the development of these materials on a larger scale.

## 1. Introduction

An appropriate definition of peloids is that enunciated by Gomes C. et al., 2013 [1]: “Peloid is a maturated mud or muddy suspension/dispersion with healing and/or cosmetic properties, composed of a complex mixture of fine-grained materials of geologic and/or biologic origins, mineral water or sea water, and commonly organic compounds from a biological metabolic activity. When the maturation takes place in the natural environment it is called natural peloid and can be considered a healing mud, and in other cases it is named peloid. According to their overall composition peloids are classified into inorganic peloids, organic peloids and mixed peloids; they also could be named medical peloids and cosmetic peloids, according to their application.”

Pelotherapy is a suitable practice for treating various illnesses and preventing skin diseases, osteoarticular disorders, and other conditions. Its applications in the cosmetic field are also significant. In a broad sense, it can contribute to improving people’s quality of life when it is used in the different spaces available for this purpose, such as spas and thermal centers [2,3].

Copahue, located in the Andes Mountains, Neuquén Province, Patagonia, Argentina, is a thermal center with abundant resources, including mineral-medicinal waters, natural peloids, artificially matured peloids, algae, and various microorganisms from the regional volcanic activity [4,5]. Due to its geographical location and elevation above sea level, the entire thermal center is covered by several meters of snow during autumn, winter, and part of spring or from May to November in the southern hemisphere. During this period, only the lagoons with acidic and hyperthermal waters remain without snow on their surface.

The closure of the Copahue Thermal Center due to inclement weather for six months has resulted in over 800 individuals losing their jobs, which is a significant social problem. The center’s remote location, 385 km from Neuquén, which contains the nearest airport, and over 1520 km from the Argentine capital, Buenos Aires, makes it difficult to access during the operating season. Therefore, it is imperative to explore alternative options for obtaining Copahue peloid-derived materials with suitable properties for pelotherapy. These materials should be easily distributable and usable even at a great distance from the center’s origin, thereby facilitating access and creating employment opportunities at the regional level.

Thermotherapy is one of the most significant benefits of peloids, as it involves maintaining the skin’s area of application at above-body-temperature levels, which leads to an increase in blood flow [6,7]. This effect is typically associated with the material’s aqueous content, and Copahue peloids exhibit the appropriate cooling kinetics for thermotherapy. The peloids’ grey color, which gives them low luminosity, also contributes to their thermal effects by absorbing radiation [8]. Patients recognize the greyish tonality as characteristic of Copahue’s peloids, making it an essential criterion of acceptability. The peloids’ composition, rich in clay minerals that provide high adsorbent properties and elemental sulfur, enables them to be classified as sulfur–peloids [4,5,9,10] with antiseptic properties that have significant dermatological applications [3,11,12].

The use of mudpack formulations, consisting of a suitable combination of cosmetic paraffin and dehydrated peloids’ solid phase, has not been evaluated in conjunction with Copahue’s peloids. This type of formulation is widely used in other countries and has resulted in renowned and long-used materials in European thermal centers, such as the Battaglia mudpack [13,14,15].

Paraffin for therapeutic and cosmetic use is a hydrophobic hydrocarbon with a low melting point, which is specifically used for pain treatment in osteoarthritis and arthritis illness [16]. Pain relief is produced when it is applied at a temperature higher than the corporal one. The mechanism by which heat acts on these processes is by increasing the extensibility of fibrous tissues such as tendons, joint capsules, and scars. It also relieves pain by relaxing secondary muscle spasms, which are frequently observed in rheumatic diseases, as well as by increasing vascular flow and endorphin production [17,18]. The pain-relieving action of heat also occurs in chronic processes: it decreases the joint stiffness seen in rheumatoid arthritis and various other movement-limiting conditions and helps to resolve inflammatory reactions and edema in their chronic phase [19,20].

Materials such as paraffin have a high energy storage capacity due to the amount of heat transferred during phase changes from the liquid state to the solid state and vice versa. For this reason, it has been studied mainly as a material that can be used in energy storage systems. In recent studies, the mixture of paraffin with clay minerals has been analyzed as suitable for such purposes [21]. To produce a material mixing paraffin and peloids (a mudpack), it is indispensable that the latter is in a dry form, free of water, so that no separation occurs between the two phases. The mudpack thus obtained can be defined as a material designed (or a peloid designed) for specific use in thermotherapy [22].

The aqueous phase represents, in the natural or matured peloids from Copahue, about 50% of the total [5]. When dehydrated through the various possible mechanisms, the water is eliminated from the peloids, resulting in a solid phase for both the insoluble clay minerals and the salts that were previously dissolved in the aqueous medium and obtaining a dry material which can be effectively mixed in the desired or appropriate proportions with paraffin [15]. A relevant aspect in obtaining a mudpack with Copahue mud is related to the presence of sulfur, which can be easily dissolved in paraffin when it reaches the liquid state [23], giving rise to a novel material that is versatile, with properties especially suitable for treatments where the presence of sulfur contributes to the curing or prevention of various ailments. This kind of material has the potential to be used in a personal form, which would allow it to be reused many times, considerably reducing the contamination that could be caused by its discard.

The aim of this work was to evaluate the potential formulation of mudpacks, through the appropriate mixture of paraffin and peloids from Copahue, as easily accessible materials for the population that is distant from the thermal center, analyzing the optimal proportions and characteristics of the resulting new materials. In particular, the thermotherapeutic properties of the designed mudpacks were evaluated.

## 2. Materials and Methods

### 2.1. Materials

The peloids used in this study were sourced from Copahue Thermal Center, located at the foot of the Copahue Volcano in Neuquén, Argentina, and specifically collected from the “Laguna Sulfurosa Madre”, which means Sulfurous Lagoon (SL) [24]. The peloids were collected in plastic bottles and were stored in a refrigerator until processing. To prepare the samples, the supernatants of the mineral-medicinal waters were separated from their corresponding solids by centrifuging the samples at 8000 rpm for 20 min using a Sorvall RC centrifuge.

### 2.2. Drying Process

The drying process involved two different methods: one part of the solid was placed in an oven at 50 °C until it reached a constant weight, while the other part was subjected to freeze drying. The freeze-drying process was carried out for 72 h using Christ beta 2–8 LD plus equipment under a pressure of 0.22 mbar, which removed water from the sample through sublimation [25]. After drying, all samples were ground and sieved through a 200 mesh, and they were then stored for further analysis. The solids obtained from each drying process were observed using an optical microscope (Nikon^®^ Eclipse 80i, Nikon, Melville, NY, USA) with different objectives ranging from 10× to 60×. To prepare the samples for observation, the dried materials were spread on a slide.

### 2.3. Mudpack Preparation

Mudpacks were prepared using a mixture of paraffin for cosmetic use (Biobellus^®^, Biobellus, Buenos Aires, Argentina) with varying percentages of previously freeze-dried peloids to prevent phase separation caused by water. The SL peloid was slowly added to liquid paraffin at 56 °C while stirring for 1 h to ensure homogenization. The concentration ratios were 0.5%, 1.5%, 3%, 6%, and 10% m/m of the dry peloid in the mudpack.

### 2.4. Characterization

All formulations were characterized by using different techniques. X-ray diffraction (XRD) analysis was performed using a SmartLab SE Rigaku^®^ X-ray diffractometer (Tokyo, Japan) to examine the crystalline structure of the mudpack. Additionally, 2θ angle scanning was conducted from 5° to 30° at a scanning speed of 2°/min.

Fourier transform infrared spectroscopy (FTIR) was used to determine the structural properties of the solids. Spectra were obtained using an IRTracer-100 spectrometer (Shimadzu, Kyoto, Japan) from 400 to 4000 cm^−1^ after preparing samples with the potassium bromide pressed disc technique, which involved mixing 3 mg of the sample with 300 mg of KBr.

Optical microscopy was employed to evaluate the redistribution of paraffin and sludge within the mudpack at various magnifications.

The color of the mixtures was determined using a portable colorimeter (Minolta Co., model CR400, Tokyo, Japan) via photocolorimetry. To standardize the measurement, a white cylindrical container was used, while the color ratios were calculated in the CIELAB uniform space. The different concentrations of mudpacks (previously melted) were placed in glass vials, and five replicates were made. The parameters L*, a*, and b* were measured, where L* represents brightness (0 = black, and 100 = white), the a* scale indicates the chromaticity axis ranging from green (−) to red (+), and the b* axis ranges from blue (−) to yellow (+) [26].

The preceding characterizations were carried out for all formulations.

### 2.5. Thermal Behavior

Cooling curves were analyzed for each of the mixtures to evaluate their thermotherapy following the method described in previous works [27]. A 50 mL polypropylene container with a screw cap and a hole in the center was used to hold approximately 28 g of the previously prepared mixture, which was then measured precisely. A temperature sensor was inserted into the hole, and the container was placed in a hot water bath at 56 °C. After reaching thermal equilibrium, the container was transferred to a thermostatic bath at 35 °C. Temperature values were recorded as a function of time using the internal NTC temperature sensor of the Testo 175 T1 ^®^ (TestoSE & Co. KGaA, Titisee, Germany).

Differential scanning calorimetry (DSC) was conducted using the TA Instruments Trios V5 to evaluate the properties of phase changes, including latent heat and melting and crystallization temperatures. The sample was heated in a nitrogen atmosphere at a rate of 10 °C·min^−1^ up to 85 °C and was then cooled down to 10 °C at the same rate.

### 2.6. Sulfur Solubility

Due to the presence of sulfur in the Copahue mud, its solubility in paraffin at 56 °C was evaluated, which was found to be 1.684 g per 100 g of paraffin [23]. To carry out the analysis, 5 g of paraffin was weighed and was placed in a 50 mL Erlenmeyer flask, which was then heated in a hot water bath between 53–58 °C with the help of a stirrer and a magnetic heater. Once the melting temperature was reached and the paraffin had become liquid, approximately 80 mg of solid sulfur with 99.96% purity was added to the flask and stirred for 2 h at the same temperature. The same procedure was repeated with the sludge sample. The solid systems obtained were characterized through XRD and DSC.

### 2.7. Reuse Tests

For the purpose of analyzing the reusability of these systems, the preservation of their thermal properties was evaluated. To assess this, we conducted a test consisting of 10 cycles of continuous heating and cooling, comparing the thermal properties of the mixture before and after the test. A 30 g sample of mudpack containing 6% mass was placed in a 50 mL polypropylene container, which was then subjected to a hot water bath at 100 °C for 15 min. The mixture was stirred gently to ensure thermal homogenization. After 15 min, the container was transferred to another water bath with a constant temperature of 20 °C for an additional 15 min. The cooling and DSC curves were analyzed to assess the thermal properties of the mixture. XRD and FTIR tests were also conducted to further characterize the mixture resulting from the 10 cycles.

## 3. Results and Discussion

Prior to being incorporated into paraffin wax, the peloid must be in a dried form to avoid phase separation in the mudpack. There are different methods of drying the peloid, such as freeze drying or oven drying, which produce solids with varying consistencies. When the peloid is freeze-dried, the resulting solid phase is easier to mill and has a softer texture, which makes it easier to handle during the mudpack preparation process. This is because the freeze-drying process modifies the texture of the clay minerals, generating aggregates with a higher proportion of macropores in the structure. In contrast, dehydration mediated by an oven or heat drying produces a more compact material that is more difficult to process further [28]. Figure 1 provides a schematic representation of the differences in layer organization resulting from each drying process. Despite these differences, the mineralogical composition of the peloid, as evaluated through XRD in previous work, remains the same regardless of the drying process used [5].

As shown in Figure 2, the particles obtained through oven drying are slightly larger in size compared to those obtained through freeze drying as observed using the optical microscopy images of the dried peloids. The small size of the freeze-dried peloids gives them a soft texture, which is a desirable characteristic for their application on the skin.

The appearance and color of the paraffin and peloid mixtures were observed to change as the proportion of each component varied. Proportions ranging from 0.5% to 10% of peloid mass in the mudpack were tested, and as smaller amounts of peloid were added, the final color of the mudpack turned grey. Figure 3 depicts the resulting mixtures’ appearance compared to the starting materials. Notably, at concentrations as low as 3% of the paraffin-embedded peloid, the resulting color closely resembles that of natural peloid rather than that of paraffin.

Figure 4 illustrates the X-ray diffraction (XRD) analysis, which shows characteristic peaks for paraffin at 7.3°, 21.4°, and 23.8° and weak peaks for the SL peloid at 11°, 15°, and 21.5°. The most representative peak for the SL peloid is located at around 23°, with additional characteristic peaks observed from 24° to 29°, which is in accordance with previous studies [4,5]. The mudpacks’ spectra show the presence of paraffin and SL mud in all the ratios, which is less evident for the 0.5% mudpack, which could be due to the low percentage of SL in its composition. The JCPDS standards (Joint Committee on Powder Diffraction Standards) for paraffin and sulfur correspond to No. 40-1995 and No. 08-0247, respectively.

Previous studies [4,5] have reported the presence of sulfur in Copahue SL mud as evidenced by peaks detected in the 21 to 28° range (see Figure 4). The feasibility of dissolving the sulfur present in the SL mud by mixing it with molten paraffin was investigated in this context. The diffractogram of pure sulfur (99.6%) exhibits prominent peaks at 11.3°, 15.2°, 22.9°, and from 24 to 28°. In contrast, the diffractogram of the paraffin–sulfur mixture indicates the presence of peaks attributed to both components. This fact suggests that the sulfur present in the mud dissolved at contact with the molten paraffin because they are nonpolar substances.

Figure 5 illustrates the IR spectra of the materials analyzed, and paraffin dominates the system. The signals detected at 729 cm^−1^ correspond to the vibration of the -CH_2_- groups of saturated hydrocarbons in linear chains. The signals at 1378 and 1472 cm^−1^ can be attributed to the presence of -CH_3_ and -CH_2_- groups, respectively. The pronounced signals below 3000 cm^−1^ are characteristic of C-H bonds without the presence of rings. The bands at 2960, 2918, and 2845 cm^−1^ correspond to the symmetric stretching vibrations of the -CH_2_- group [29], indicating that the structure of this commercial paraffin for cosmetic use is aliphatic with single bonds. In contrast, the IR spectrum of the peloid SL shows the characteristic signals of Si-O stretching vibrations around 1041 cm^−1^. The deformation of the internal OH groups is observed at 910 cm^−1^, and the two signals at 534 and 468 cm^−1^ are associated with the deformations of the Al-O-Si and Si-O-Si bonds, respectively, as reported in the literature for kaolinites [30], one of the predominant minerals in these systems. For the mud, the weak band observed at 3695 cm^−1^ is attributed to the stretching of the inner surface hydroxyls between the adjacent layer, while the band at 3622 cm^−1^ corresponds to the stretching of the hydroxyls of the inner octahedral layer [31].

The dominance of the signals inherent to paraffin can be attributed to its preponderance in the mixtures. The peloid, on the other hand, is undetectable in mixtures with lower quantities but can be detected by the signal at 1041 cm^−1^ in mixtures containing 6% and 10% peloid. Figure 5 shows that when paraffin is mixed with the peloid, the absorption peaks remain unaltered compared to the initial peloid and paraffin. This suggests that there is no chemical interaction between the two compounds, and no new absorption band appears in the spectrum [32].

Beyond the visual aspect, color is a fundamental characteristic of this kind or type of material. It is important for both patient acceptability and the ability to absorb electromagnetic radiation, which contributes to enhancing its thermotherapeutic properties. Figure 6 displays the lightness parameters for these materials, with the highest values observed for both pure paraffin and the dry peloid. The high L* values indicate that these materials have a higher capacity to reflect light. In materials that are intended for use in thermotherapy, it is favorable to have a low L* value associated with a high capacity to absorb the electromagnetic radiation in the visible field. Mudpack formulations containing 3% to 10% SL exhibit lower luminosity values than the wet peloid, and they even exhibit values that are considerably lower than the values of the original materials, paraffin and the dry peloid, when taken separately. Thus, formulations achieved with concentrations of 6% and 10% of peloid are more suitable. Their chromaticity values, as shown in Figure 6, are well-differentiated from those associated with paraffin, with the mudpacks located in the middle zone.

To complete the structural analysis of the mudpack and its precursors (paraffin and the peloid), samples were observed under an optical microscope. The results are shown in Figure 7, where darker areas corresponding to the color of the mudpack are observed with increasing concentration. When both components interact, a new structure with a relatively homogeneous appearance emerges. Paraffin has a crystalline structure, which can be determined from the XRD and is reflected in the organized distribution observed in the microscope image.

One of the most relevant aspects of peloids and paraffin waxes is their thermal properties. Figure 8 displays the cooling curves obtained for each of the analyzed materials, representing the temperature decrease process of the systems when they are inside a thermostatic bath at the same temperature as body temperature. Natural Copahue peloids have good thermotherapeutic properties, and paraffin is used not only in the cosmetic field but also widely in energy storage devices. Its advantages, such as high latent heat, wide thermal stability, and low cost, have been widely studied [21,32,33]. Furthermore, mudpacks are applied as thermotherapeutic agents in the medical field.

The cooling rate, which is determined by the ratio of the temperature variation (∆T) with respect to the time variation (∆t) in each interval of the experimental curve, was calculated to be 0.6 °C·min^−1^ for both paraffin and the mudpack. However, for the peloid, higher values ranging from 0.6 to 4.8 °C·min^−1^ were obtained. This suggests that mudpack mixtures have a slower heat release, which is a desirable property for their use as a thermotherapeutic agent. Mudpacks with different proportions of the peloid showed that their thermal behavior tended to remain like that of paraffin, with a higher cooling rate at higher concentrations of the added peloid. Pure paraffin and the mudpacks at 0.5 and 1.5% had similar behaviors, while the formulations at 3 and 6% formed another group with higher cooling speeds. The mudpack at 10% had the highest cooling speed among all those tested in this study. Nonetheless, in relation to their thermal properties, all the mudpacks were better than the natural peloid. This is because, in the conventional time of application (usually 20 min), the temperature values of all the mudpacks were still several degrees above the 37 °C of the skin. All the mudpacks presented excellent thermal properties to achieve the desired vasodilatation effect associated with temperature values higher than the baseline body temperature. The DSC technique was used to characterize the phase change properties of the obtained paraffin/peloid composite. Figure 9 shows the DSC curves. The heating process of the materials is indicated by the continuous lines while the cooling process by the dashed ones. Table 1 provides the corresponding parameters.

The responsible use of these materials requires environmental consideration, including minimizing the waste generated during pelotherapy treatments. The possibility of reusing the mudpack multiple times by the same patient in a domestic setting was evaluated. Thermal stability was assessed by conducting 10 heating and cooling cycles on the 6% sample, and the cooling curve of the final sample was analyzed. The results, presented in Figure 10, demonstrate that the thermal properties improved even after 10 melting and solidification cycles, with a decrease in the cooling rate. This suggests that each cycle improves the cooling rate. This could be attributed to the fact that the homogenization between paraffin and the peloid would improve with each cycle, but further studies are needed to corroborate this assumption.

The stability of the mudpack was evaluated after ten heating–cooling cycles through XRD, DSC, and FTIR tests. The results are shown in Figure 11a,b and Figure 12. No significant differences in peak values were observed compared to the initial sample’s diffractograms, indicating that the structural and chemical properties were conserved.

Using DSC curves mudpacks were analyzed for their melting (heating process) and crystallization (cooling process) enthalpy, and their related temperatures, represented by ∆H_m_, ∆H_c_, T_m_ and T_c_, respectively. The results showed that the T_c_ and T_m_ values of the mudpacks were like those of paraffin wax, indicating that there was no reaction between paraffin and the peloid during the mixing process. A slightly higher value was obtained for the mudpack after the heating–cooling process. When comparing the ∆H_m_ and ∆H_c_ values, a significant difference was observed between the original sample and the sample (6%) corresponding to the heating cycles. This could be attributed to the fact that the cycling process resulted in a more homogeneous system. Notably, both the 10% mudpack and the recycled mudpack showed higher latent heat values than paraffin.

## 4. Conclusions

Copahue peloids, rich in sulfur and with a characteristic grey color, are accessible to patients only during the months of December to May due to climatic reasons. The formulations of paraffin and Copahue peloids result in materials (named mudpacks) with the potential to be used in a personal way anywhere and at any time of the year.

The presence of paraffin in the mudpack guarantees thermotherapeutic properties with a longer duration during application. In addition to the thermal properties provided by paraffin, the incorporation of the Copahue peloid provides sulfur and clay minerals, thus adding disinfectant, keratolytic, antiseptic, and adsorbent properties.

The color obtained in the mudpacks contributes to a higher absorption of electromagnetic radiation and increases patient acceptability.

The mudpacks obtained were characterized through the XRD, IR and DSC techniques. The cooling curves allowed us to establish the criteria required to select the most suitable concentration of the peloid in the formulation: 6% m/m.

Mudpacks have exceptional thermal properties that allow for very low-speed cooling kinetics due to the presence of paraffin in the system. The ability to slowly release heat is one of the most important properties of mudpack applications, allowing them to be used as thermotherapy agents that can be easily applied to the patient’s skin.

The stability of the thermal properties and the chemical structure of the mudpack were maintained after the thermal cycles, evidencing that the mixture does not undergo changes in its crystalline structure and chemical bonds after thermal treatment. Finally, this shows the high stability of the mudpacks, allowing their reuse and representing an environmental advantage.

## Figures and Tables

**Figure 1 materials-16-05062-f001:**
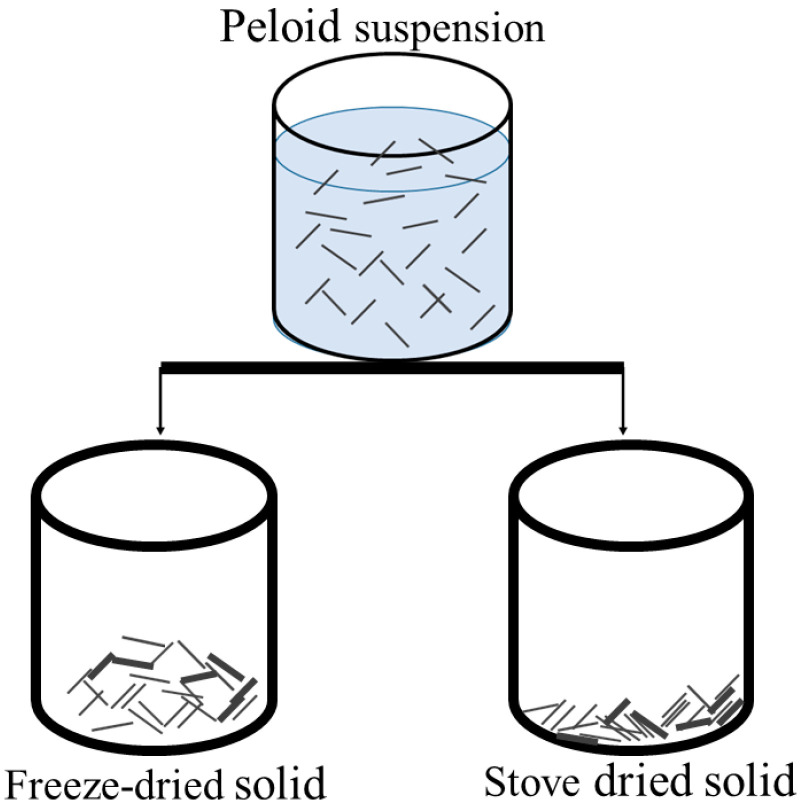
Scheme of the organization of sheets according to the drying method.

**Figure 2 materials-16-05062-f002:**
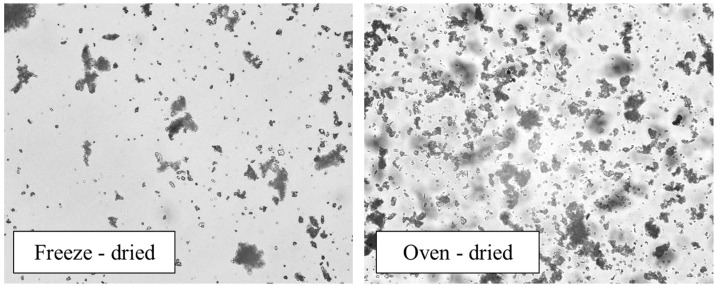
Microscopy of freeze-dried and oven-dried peloid.

**Figure 3 materials-16-05062-f003:**
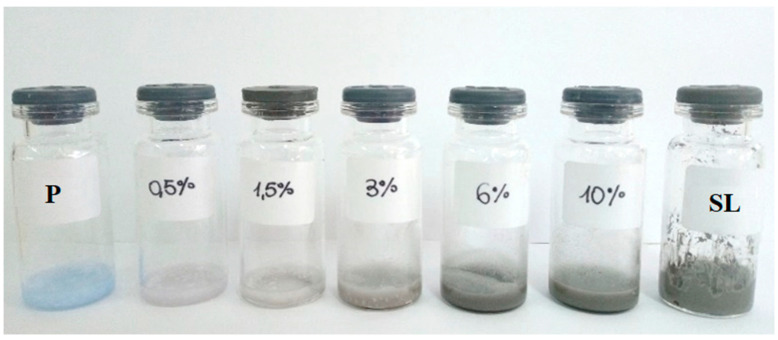
Paraffin (P) and mudpacks with different concentrations of peloid (SL).

**Figure 4 materials-16-05062-f004:**
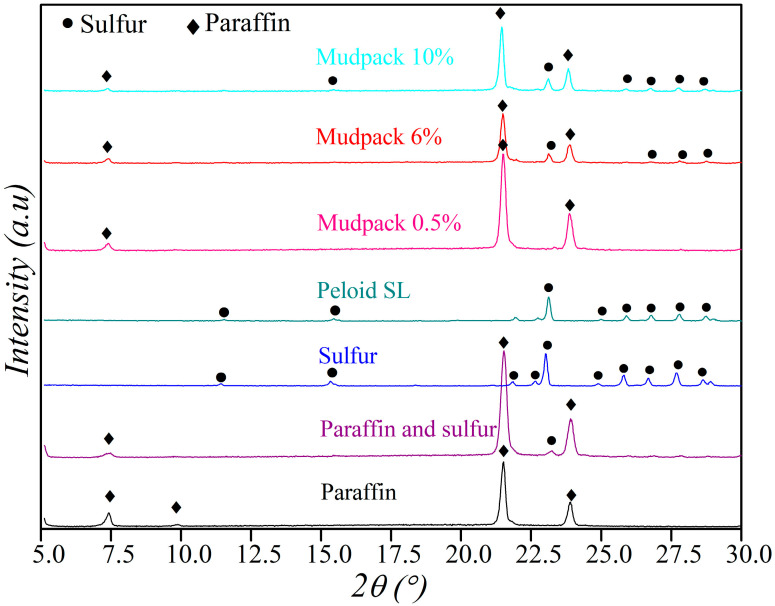
X-ray diffractograms (XRD) of paraffin, mixture of the paraffin and sulfur, peloid, sulfur, and mudpacks.

**Figure 5 materials-16-05062-f005:**
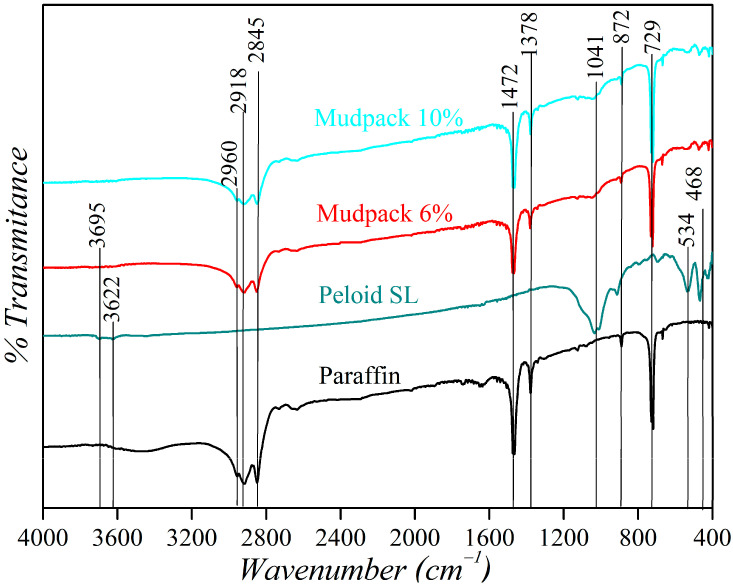
Characteristic signals in the IR spectra of the mudpack 10%, mudpack 6%, peloid SL, and paraffin.

**Figure 6 materials-16-05062-f006:**
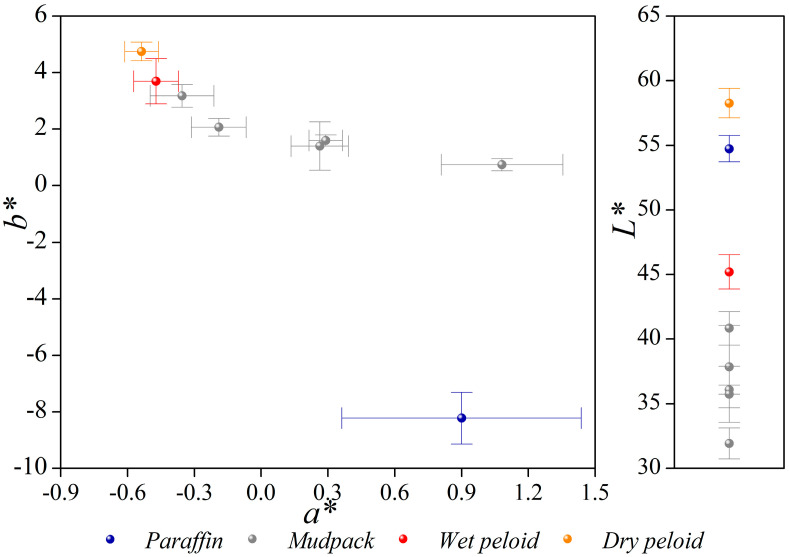
Color parameters of the samples.

**Figure 7 materials-16-05062-f007:**
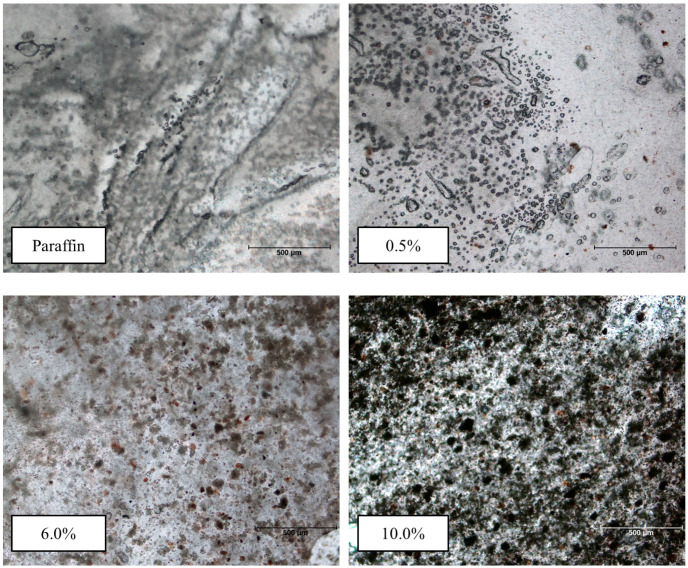
Optical microscope images (4×) obtained for mudpacks of different concentrations (%).

**Figure 8 materials-16-05062-f008:**
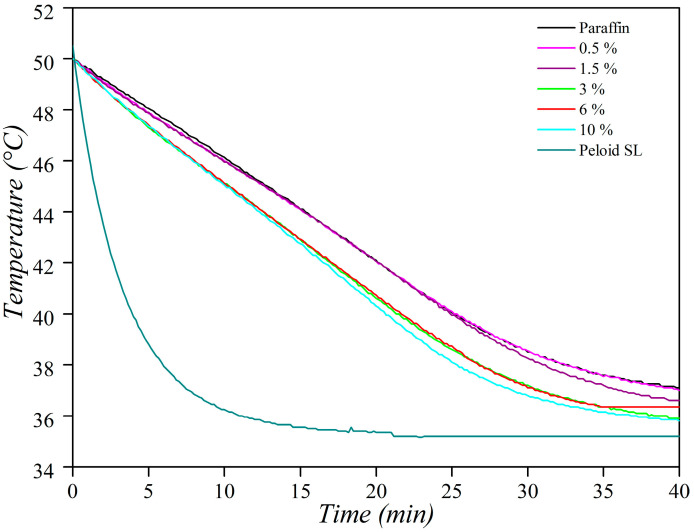
Cooling curves of the mudpacks, paraffin, and peloid.

**Figure 9 materials-16-05062-f009:**
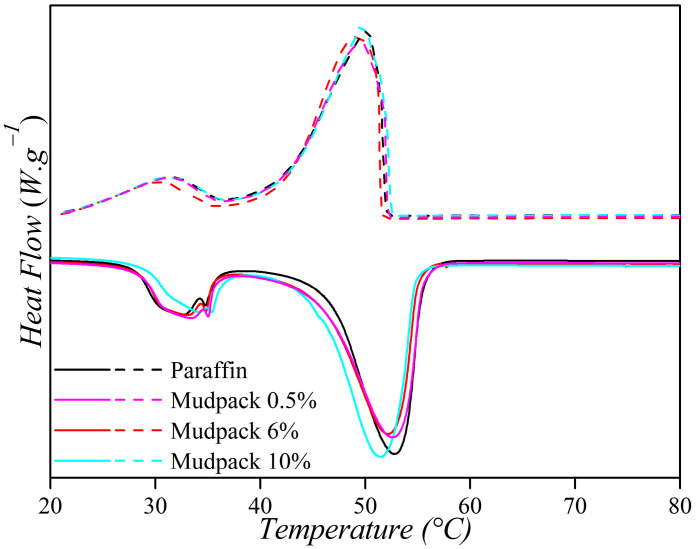
DSC paraffin and mudpacks.

**Figure 10 materials-16-05062-f010:**
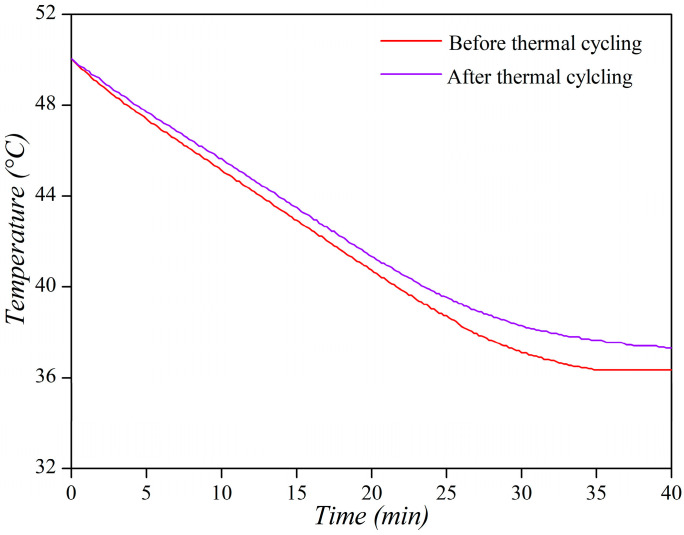
Cooling curves of the material initially and after the 10 reuse cycles.

**Figure 11 materials-16-05062-f011:**
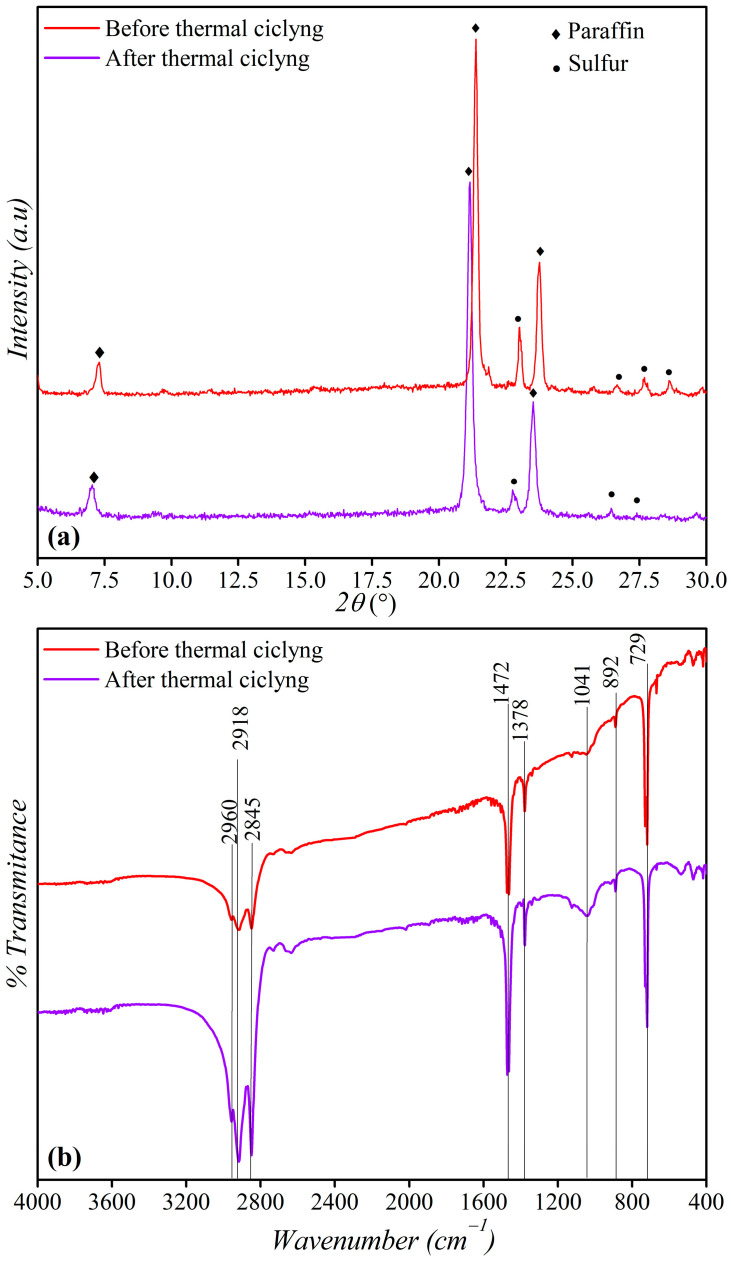
(**a**) XRD and (**b**) IR spectra of the 6% original and reuse.

**Figure 12 materials-16-05062-f012:**
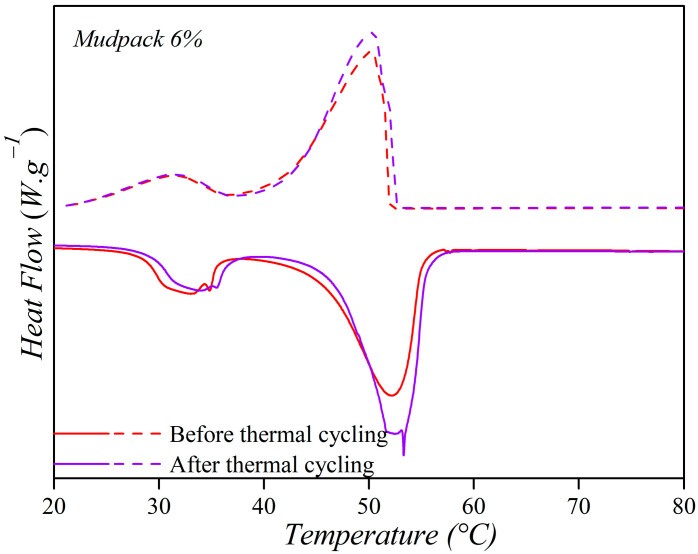
DSC of the initial material and then of 10 heating and cooling cycles.

**Table 1 materials-16-05062-t001:** Latent heat storage properties of mudpacks.

Sample	$∆H_{m}$ (J/g)	T_m_ (°C)	$∆H_{c}$ (J/g)	T_c_ (°C)
Paraffin	135.16	52.90	134.79	49.24
Mudpack 6%	125.34	52.32	128.77	49.97
Mudpack 10%	140.91	51.48	135.13	49.45
Mudpack 6% 10 cycles	147.41	53.33	146.58	50.48

## Data Availability

Data will be made available on request.

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
