# Peer review of "Paraffin–Peloid Formulations from Copahue: Processing, Characterization, and Application"

_materials, 2023, doi:10.3390/ma16145062_

Round 1

Reviewer 1 Report

Please provide statistical analysis.

The study focuses on the mudpack formulation based on the addition of SL peloid (0.5, 1.5, 3, 6, and 10%). However, the analyses were conducted partially. Not all formulas were characterized using all (or at least most) characterization. This may lead to scientific flaws.

L138. Section 2.3 Provide information on the amount of paraffin the authors used

L142. What do the percentages mean? Do  they refer to weight percentage? Ratio mass of SL to the ratio of the liquid paraffin? How long was the homogenization? Please provide clear information.

The authors need to provide justification for the usage of oven vs freeze drying. Optical microscopy has been provided, but no further discussion. The authors shall discuss the effect of different types of drying methods.

Figure 1 is not that important. It can be deleted.

The resolution of the Figures is too low, please improve.

L231-L236. Please provide the JCPDS number that the authors referred.

Figure 4. Please carefully do the stacking of the diffractogram. The current version has low readability.

Figure 4. The diffractogram of Paraffin _ Sulfur seems to have an extremely high intensity at 2tetha of ca 21 deg. Can you explain this?

Figure 5. Please arrange the label according to the sequence of the signal. Start with mudpack 10% and end with Paraffin.

Author Response

Dear Reviewer,

                We would like to express our gratitude for your valuable feedback and insightful suggestions regarding our manuscript titled “Paraffin-Peloid formulations from Copahue: processing, characterization and application”. We have carefully reviewed your comments and have made the necessary revisions to address each point raised.

                In this response letter, we provide a detailed account of the changes we have made in accordance with your suggestions. We have thoroughly revised the manuscript to enhance its clarity, scientific rigor, and overall quality. Below, we outline our responses to each specific comment, indicating the modifications we have implemented.

                We have carefully considered your recommendations and believe that our revisions have improved the manuscript. We hope that the revised version adequately addresses your concerns and meets your expectations.

                Thank you for the opportunity to revise and resubmit our manuscript. We look forward to your further guidance and input during the next stages of the review process.

Reviewer 2 Report

In their work, Sanchez and co-authors present and analyse mudpacks made of paraffin and Copahue peloids. These mudpacks are formulated in order to give accessibility to peloid treatments all year round and also in locations far from thermal centers.

The mudpacks retain or even improves the main features of peloids in particular regarding the slow release of heat ideal for thermotherapy.

The work, on the overall is interesting. However, I have one main concern about its usefulness. In fact, according to what I can see in Figure 8 and what is written in the text (line 314-315), the cooling curve of pure paraffin is comparable to that of mudpacks 0.5%n and 1.5%, while the cooling speed disadvantageously increases for higher peloid contents. My question is: what is the advantage of using a mudpack rather than pure paraffin (or pure paraffin with the addition of some sulphur if antiseptic properties are required)?

In addition, I have some minor comments/suggestions.

-          Line 84: I am not sure that the sentence “at a higher than corporal temperature” is correct. I would rather write: “at a temperature higher than the corporal one”

-          Line 97: change “indispensable that the latter be in a dry form” into “indispensable that the latter is in a dry form” or “indispensable for the latter to be in a dry form

-          Line 111: replace “discarding” by “discard”

-          Line 140: the mudpacks were prepared with the freeze-dried peloid only? Or where there also mudpacks made with the oven-dried peloid? Please explain it better in the text (clarifying, if it is the case, the choice of using only the freeze-dried peloid for the mudpacks).

-          Line 209: I imagine “DRX” stands for “XRD”

-          Lines 217-219: the need of removing water was already clearly mentioned at the beginning of the paragraph (line 199-200); no need to repeat it.

-          All figures are blurry and legends are difficult to read. Please, provide figures with higher definition and write legends in a bigger size.

-          Lines 245-247: “The low concentration of sulphur…molten paraffin” this sentence is not clear to me. I do not understand what you mean. Sulfur is soluble in paraffin, this is clear. What is the point about low concentration, though?

-          Lines 331-333: do you have an explanation for the improvement of the thermal properties induced by repeated cycles?

-          Line 348: isn’t it more correct to write “melting and crystallization enthalpy” instead of “melting and crystallization properties”?

Minor corrections are required (see comments in the previous section)

Author Response

(The authors gave the same response as above.)
